# Initial Imaging Findings of Breast Liposarcoma: A Case Report

**DOI:** 10.3390/diagnostics13142428

**Published:** 2023-07-20

**Authors:** Sharifa Khalid Alduraibi

**Affiliations:** Department of Radiology, College of Medicine, Qassim University, Buraidah 52571, Saudi Arabia; salduraibi@qu.edu.sa; Tel.: +966-50-5173364

**Keywords:** liposarcoma, breast cancer, phyllodes tumor, pleomorphic liposarcoma, mammogram

## Abstract

Liposarcoma of the breast is a rare form of cancerous tumor that can be mistaken for primary breast cancer. A recent instance involved a woman who was 54 years old and went in for her annual screening mammogram. The mammogram revealed that she had a 1 cm focal asymmetry of equal density in her right axillary tail, approximately 9 cm from the nipple. After nine months, the patient observed a rapidly growing mass even though the initial ultrasound scan did not detect anything unusual. A targeted mammogram demonstrated a large and dense mass confined to the right axillary tail, followed by an ultrasound scan that revealed a heterogeneous hyperechoic, echogenic mass. Histopathology after surgery showed that the patient had an undifferentiated pleomorphic breast liposarcoma. This diagnosis was reached after the patient underwent surgery.Liposarcoma of the breast is a concerning condition that needs careful management and close monitoring, although it is relatively uncommon. Early detection of the patient’s condition and prompt treatment can help improve the patient’s prognosis. This can be accomplished by remaining vigilant with routine screenings and following up on any unusual findings or changes in breast tissue. However, it is possible to diagnose this condition as primary breast cancer incorrectly; consequently, healthcare providers need to conduct comprehensive evaluations to ensure diagnostic accuracy and the delivery of appropriate treatment.

## 1. Introduction

Breast sarcomas are occasionally mistaken for other types of primary breast cancer. Breast sarcomas are extremely uncommon, and their clinical presentation is quite similar to that of other types of primary breast cancer. Less than 1% of all cancerous breast tumors are considered to be sarcomas of the breast [1]. A malignant soft tissue tumor originating in lipocyte cells is known as the primary pleomorphic liposarcoma of the breast. This form of the disease is extremely rare. It is one of the rarest types of breast cancer, accounting for only 0.3% of all malignant breast tumors [2,3]. Breast liposarcomas are a type of breast sarcoma that affects women in their fourth to sixth decades. They account for 3–24% of all breast sarcomas [3].

It is common for breast liposarcoma to manifest as a noncancerous lump that can either gradually or swiftly expand. Imaging tests, including mammography, ultrasonography, and/or magnetic resonance imaging (MRI), are commonly used to diagnose this condition. Imaging can be used to diagnose breast liposarcoma; however, this can be difficult because the tumor might look like other types of breast tumors, both benign and malignant. In addition, to confirm the diagnosis, an ultrasound-guided core needle biopsy is performed by a radiologist, and pathology testing and/or genetic analysis are mandatory.

Surgery is the primary method of treatment for breast liposarcoma, and total surgical excision is considered to be the best possible outcome. Adjuvant therapy may consist of radiation therapy in cases of large masses or where total resection is not feasible, but chemotherapy and hormonal therapy have no role [3,4]. The overall prognosis for patients with breast liposarcoma can be unpredictable, despite the treatments that are currently available, and certain patients have a significant risk of recurrence and metastasis. The well-differentiated type accounts for 90% of cases, while the pleomorphic and round cell types make up 20% of cases with higher grades [1].

The inadequate understanding of the pathogenesis of breast liposarcoma is one of the hurdles that must be overcome to understand and treat the disease. A number of genetic abnormalities have been identified in breast liposarcoma, including amplification of MDM2 and MDM4 inhibitors in combination with ER (Estrogen Receptor) degraders or CDK4/6 inhibitors, which have garnered considerable interest as potential therapeutic approaches [5]. Although the identification of biomarkers might assist in diagnosing and treating breast liposarcoma, researchers still need to gain a deeper understanding of the genetic mechanisms underlying the disease.

In summary, breast liposarcoma is an extremely uncommon form of breast cancer that is notoriously difficult to diagnose and treat. Its pathogenesis is not fully known, and its prognosis is unpredictable. Despite treatment, some patients have a high risk of cancer returning or spreading to other parts of the body. Because our knowledge of this condition is so poor, it is essential that additional investigations be conducted into the disease’s diagnosis, therapy, and pathophysiology.

To the best of our knowledge, there are currently no published papers documenting the early findings of breast liposarcomas on mammograms during the course of our research.

## 2. Case Presentation

A 54-year-old woman came to our facility for the screening procedure. A mammogram was performed on her. The patient had no remarkable medical or family history, and she was here for routine screening. Mammography showed focal asymmetry of equal density in the upper-outer quadrant of the right breast, which was 9 cm distant from the nipple. The focal asymmetry was 1 cm in size. This finding partially persisted in the additional targeted compression view (Figure 1). The targeted ultrasound examination did not reveal any findings that were suitable for inclusion in the criteria. After an additional nine months had elapsed, a breast mass in the axillary region of the right breast was identified as being the source of her symptoms. During the clinical examination, it was discovered that the patient had a mass that was uncomfortable, moveable, and irregularly shaped and that measured 2 × 2 cm^2^.

There was no indication of inflammation or swelling in either breast. A mammogram showed a large fat-containing oval-shaped lump with a circumscribed outline, and the mass in the upper outer quadrant of the right breast was approximately 9 cm from the nipple and measured 3 × 3.5 cm^2^ (Figure 2). However, mammography of the left breast showed nothing out of the ordinary. The subsequently targeted ultrasonography (US) revealed a heterogeneous isoechoic irregularly shaped solid mass that measured 3.3 × 3 cm^2^ and had eccentric cystic foci placed at the 9 o’clock position and approximately 8 cm away from the nipple (Figure 3). Differential diagnoses include phyllode tumor (PT) in addition to liposarcoma. The patient sought treatment at a tertiary hospital, where she underwent a right breast biopsy, and based on the findings of that procedure, a right lumpectomy without an axillary dissection was conducted. When seen under a microscope, the tumor was found to contain mature adipocytes, atypical spindle cells, and multivacuolated lipoblasts. All of these cell types were discovered to be embedded in a loose myxoid to fibrous stroma within the tumor itself. This information is provided by the pathology report. In addition, there were a significant number of regions that had pleomorphic cells, multinucleated, weird, enormous cells, and lipoblasts (Figure 4 and Figure 5).

In the case that we are currently presenting, the patient is a 54-year-old woman who underwent surgical intervention but did not receive any additional treatment, such as chemotherapy or radiotherapy, at the time of the initial diagnosis. Further guidelines for follow-up care include routine clinical checks as well as mammography and breast sonography. The surgical margin was negative. Both radiotherapy and chemotherapy were eliminated from consideration as possible therapies for the condition.

## 3. Discussion

Breast cancer is a common type of cancer that affects many women worldwide. However, there is a rare form of breast cancer known as primary breast sarcoma. These are malignant tumors that develop from the mesenchymal tissue of the mammary gland. Sarcomas are an infrequent occurrence in the breast, constituting a mere 1% of all breast malignancies, with their occurrence primarily observed in the retroperitoneum and the extremities [6]. Despite being a type of breast cancer, primary breast liposarcoma accounts for approximately 0.003% of all malignant breast tumors. This means that they are not commonly seen in clinical practice [6].

Nonetheless, it is essential to be aware of this type of cancer, as early detection and treatment can improve the chances of survival [7]. Patients typically present in the clinical setting with a breast tumor that is localized to one side only and is growing at a rapid rate. In certain instances, the size of the tumors has been documented as reaching as much as 40 cm [8]. Liposarcoma, a type of malignant tumor, is a rare occurrence. It can manifest as either a pure liposarcoma or develop within a phyllodes tumor (PT) [9].

First, it is essential to emphasize the significance of imaging modalities in the process of diagnosing breast liposarcoma. Mammography, ultrasound, and magnetic resonance imaging (MRI) are just some of the diagnostic tools that can be used to determine whether or not a tumor is present and how extensive it is. In a study that was conducted not too long ago, an MRI examination demonstrated a sensitivity of 100% and a specificity of 83% [10].

The use of imaging to distinguish breast liposarcoma from other types of tumors, such as lipoma, which can present similarly to the signs and symptoms of breast liposarcoma, is another potential topic of discussion. In a recent study conducted by Nagarajan et al., various imaging techniques were utilized to identify the distinguishing features of primary breast liposarcoma when compared to other benign or malignant breast tumors. The researchers focused on key elements such as the fat component, the presence of solid or soft tissue, and internal vascularity using mammograms and ultrasound imaging. By analyzing these characteristics, the team aimed to establish a reliable method for differentiating primary breast liposarcoma from other breast tumors [10].

Imaging can also help doctors determine the best course of treatment by providing a detailed picture of the tumor and how far it has spread. This can help them decide whether surgical excision, radiation therapy, or chemotherapy is the best option. Imaging can also play an important part in monitoring the progression of treatment and confirming the treatment’s effectiveness.

The rarity of breast liposarcoma is one of the primary obstacles in the imaging diagnosis of the disease, as this can make it difficult to diagnose the disease early and in a timely manner. However, recent research indicates that imaging plays an essential role in facilitating early detection. To accurately diagnose these rare tumors, imaging makes use of innovative techniques such as ^18^F-fluorodeoxyglucose-positron emission tomography (PET)-CT, which has emerged as a dependable diagnostic tool for differentiating between well-differentiated liposarcoma (WDLPS) and dedifferentiated liposarcoma (DDLPS). A recent study has shown that PET-CT exhibits a sensitivity of 83.3% and a specificity of 85.7% in accurately identifying these two subtypes of liposarcoma, which led to more prompt treatment [10].

In regard to mammography, tumors that exhibit a combination of fat and soft tissue with an unclear outline tend to appear quite frequently. However, mammograms usually depict dense masses that have a well-defined shape [6]. It is worth noting that cases of primary liposarcoma of the breast, which is a relatively rare condition, are not often documented in imaging results. Imaging methods have limitations in regard to providing accurate diagnoses, and as a result, the final diagnosis often relies on surgical resection and subsequent pathological examination [11].

On breast ultrasonography, these lesions look primarily echogenic, with solid regions and vascularity within. MRI assists in the characterization of the fat and soft tissue components and the detection of the exact degree and existence of axillary lymphadenopathy. On contrast-enhanced MRI, these lesions appear as enhancing masses with early peak contrast enhancement and slow fading or plateauing of the contrast enhancement on kinetic analysis. The combination of (FDG) PET-CT is able to assess metabolic activity within the primary lesion, as well as the existence of both local and distant metastases. On (FDG) PET scans, they also exhibit avid uptake [2]. The mammography findings are shown in the current scenario as partially persistent focal asymmetry with no underlying ultrasound findings, and it is worth performing a breast MRI for further clarification instead of performing a short follow-up to prevent aggressive progression.

The therapy and prognosis depend heavily on histological distinction, grade, and type. Well-differentiated liposarcomas, also known as atypical lipomatous tumors, are the most prevalent type of liposarcoma, followed by myxoid/round-cell liposarcoma (MLP), pleomorphic liposarcomas (PLPs), and dedifferentiated liposarcomas (DLPs) [6]. Currently, liposarcomas are classified into four main categories based on their pathological characteristics. Even if there is a large degree of variance comparable to that discovered during the clinical examination, imaging investigations demonstrate that distinct liposarcoma subtypes share certain characteristics. This is the case even though there are many different subtypes. Malignant PT is quite prevalent when associated with well-differentiated liposarcoma (WDL) or dedifferentiated liposarcoma (DDL), although it is uncommon when associated with MLP and PLP [8]. Despite this, it is extremely important for patients who have WDL to have a long-term follow-up because there is a risk of local recurrence as well as delayed dedifferentiation [12,13].

Imaging is an essential aspect of the diagnostic process for breast liposarcoma, a very rare form of cancer, and imaging is vital for both discovering and treating the condition. This is because breast liposarcoma cannot be detected without imaging. If clinicians make use of a wide array of imaging modalities and keep themselves up to date on the most recent research, they will be able to deliver more accurate diagnoses and more successful therapies for their patients.

## 4. Conclusions

Liposarcoma of the breast is an extremely uncommon form of cancerous tumor that begins in the fatty tissue of the breast and can sometimes spread to other parts of the body. It is frequently challenging to diagnose because it may initially manifest as focal asymmetry that lacks specificity. This is one reason why it is so common. Radiologists who are attempting to ascertain the type of breast lesion may face difficulties in their diagnostic work as a result.

It is extremely important for radiologists to be aware of the fact that breast liposarcoma can manifest itself in this manner. This awareness can make a substantial difference in accurately diagnosing the problem in a timely way and enabling prompt therapy to be administered. A better prognosis for patients can also be ensured by early identification.

As a result, a close follow-up examination with mammography is strongly recommended by radiologists. If a mass is found during the initial exam, but the appearance of the mass is suspicious or if it does not appear to be breast cancer, then a follow-up exam can be especially advantageous. This can assist in ruling out other possible illnesses and allowing for a more accurate diagnosis to be made.

In conclusion, breast liposarcoma must be detected at an early stage for patients to receive treatment that is both prompt and accurate. Radiologists need to be aware of the fact that it can manifest as focal asymmetry in a localized area that lacks specificity. Mammograms should be part of a patient’s routine follow-up care because they can play a vital role in ensuring an accurate diagnosis.

## Figures and Tables

**Figure 1 diagnostics-13-02428-f001:**
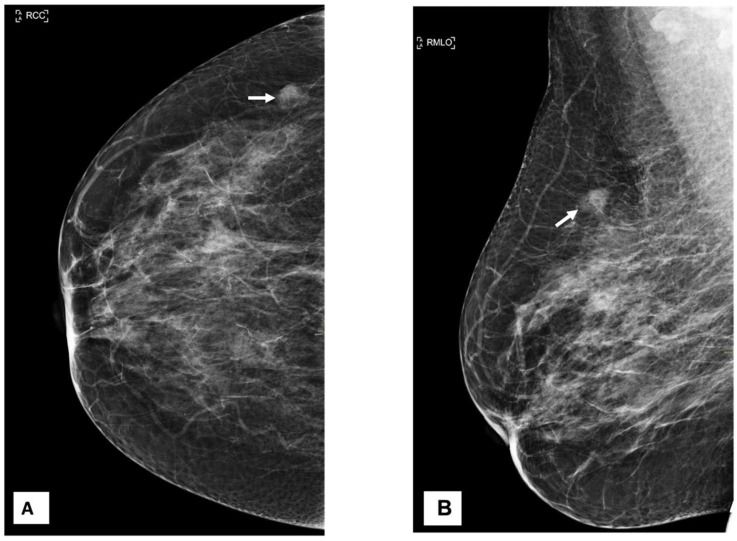
Craniocaudal (CC), mediolateral oblique (MLO), and MLO compression view mammography. (**A**–**C**) show a partial persistent oval-shaped fat containing 1 cm of focal asymmetry in the upper-outer quadrant of the right breast (arrows).

**Figure 2 diagnostics-13-02428-f002:**
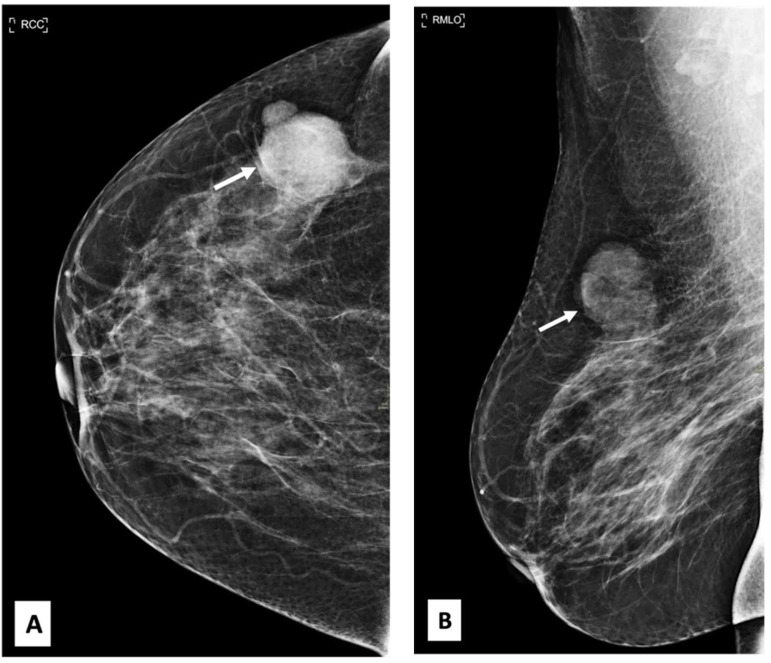
Craniocaudal and mediolateral oblique nine-month follow-up mammography of the right breast. (**A**,**B**) A large, circumscribed fat-containing oval-shaped lump approximately 9 cm from the nipple measuring 3 × 3.5 cm^2^ (arrows) with no associated suspicious features (suspicious microcalcification or architectural distortion).

**Figure 3 diagnostics-13-02428-f003:**
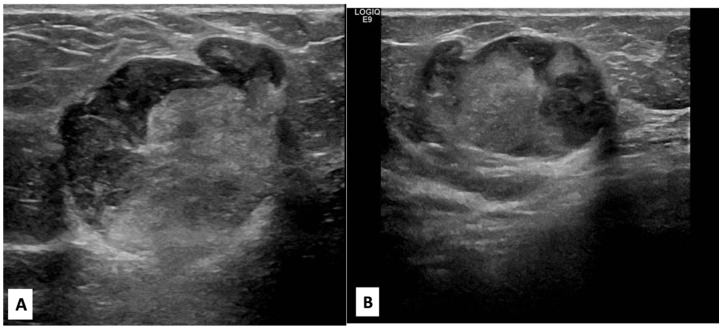
Radial, anti-radial, and color Doppler right breast targeted ultrasound scan images. (**A**,**B**) shown at the 9 o’clock position and 8 cm from the nipple is a 3 × 3.3 cm^2^ superficial circumscribed irregular shape of a heterogeneous isoechoic mass with a small peripheral cystic component perpendicular to the skin; no posterior shadow; and no associated suspicious features (suspicious microcalcification or architectural distortion). (**C**) Color Doppler image of the mass showed peripheral vascularity.

**Figure 4 diagnostics-13-02428-f004:**
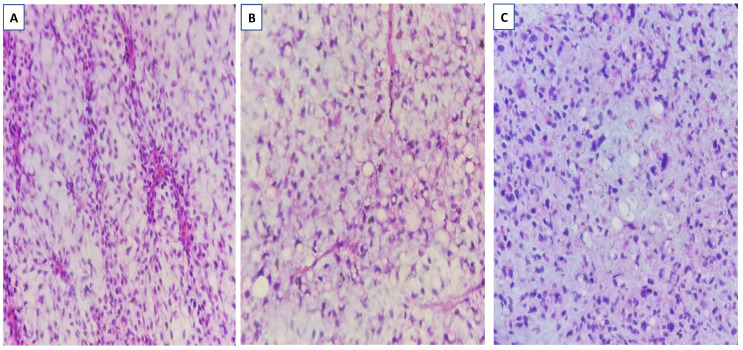
Liposarcoma with mature adipocytes, atypical spindle cells, and multivacuolated lipoblasts embedded in a loose myxoid to fibrous stroma (**A**,**B**) and areas with a disorderly pattern, pleomorphic cells, multinucleated bizarre giant cells, and lipoblasts (**C**). Original magnification is 400×.

**Figure 5 diagnostics-13-02428-f005:**
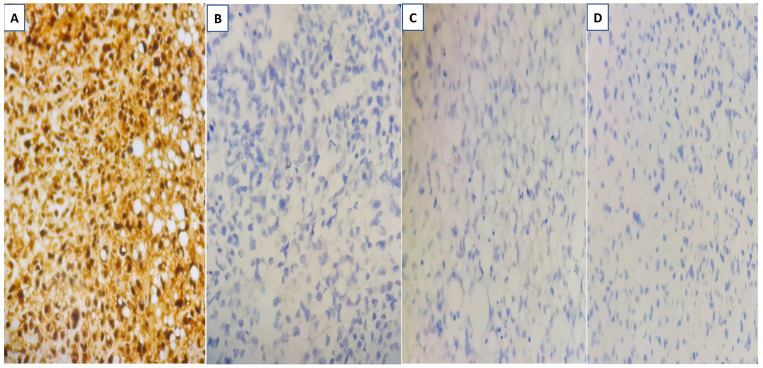
Liposarcoma with strong diffuse positive S100 immunohistochemical staining (**A**), negative expression of CKAE1/AE3 (**B**), P63 (**C**), and negative HMB-45 expression (**D**). Original magnification is 400×.

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
