# Peer review of "Initial Imaging Findings of Breast Liposarcoma: A Case Report"

_diagnostics, 2023, doi:10.3390/diagnostics13142428_

Round 1
Reviewer 1 Report
English is not excellent so I suggest that the text be reviewed by a mother-tongue English teacher.
There are also some typing errors that must be corrected, for example, in line 20 dot and comma must be replaced with dot; in line 82, "however" must be written in uppercase; in line 176 p<. 05 should be corrected to p<0.05.
Rows 73, 81, 83 and 94: Figure 1, Figures 2, Figures 3 and Figure 4-5 should be written in brackets.
In addition, in the line 197, the name of the radiopharmaceutical used for PET-CT should be added.
English is not excellent so I suggest that the text be reviewed by a mother-tongue English teacher.
Author Response
Comments and Suggestions for Authors:
(Reviewer-1)
English is not excellent so I suggest that the text be reviewed by a mother-tongue English teacher.
There are also some typing errors that must be corrected, for example, in line 20 dot and comma must be replaced with dot; in line 82, "however" must be written in uppercase; in line 176 p<. 05 should be corrected to p<0.05.
Rows 73, 81, 83 and 94: Figure 1, Figures 2, Figures 3 and Figure 4-5 should be written in brackets.
In addition, in the line 197, the name of the radiopharmaceutical used for PET-CT should be added.
Response 1
All the above-mentioned comments have been addressed and rectified.
Reviewer 2 Report
The imaging to perform diagnosis of breast stromal neoplasms is only indicative. The only way to distinguish a stromal lesion from a benign or malignant epithelial tumor or biphasic neoplasm (such as fibroadenomas o phylloid tumor) is the histological examination used needle core biopsy.
Even histologically, it is sometimes difficult to distinguish a lipoma from a well-differentiated liposarcoma. Pure primary breast sarcoma that does not originate from the malignant component of a phyllodes tumor or from the differentiation of a metaplastic carcinoma (principal differential diagnosis to considered) is extremely rare. Only with a total macroscopic sampling of neoplasm this diagnosis could be pose after the histological esclusion of epithelial neoplastic component.
I don't understand what is meant by "Pure primary liposarcomas or osteosarcoma-phyllodes are the two possible origins of osteosarcoma-phyllodes in breast liposarcomas (167-168)". It is a non-existent entity in WHO.
However, I agree that radiologists must know the extent of liposarcoma and in general of all potential stromal neoplasms of the breast and that radiology, after biopsy histological diagnosis, is useful in planning surgery and treatment.
However, radiology alone cannot diagnose rare stromal neoplasms of the breast.
Furthermore, I would have put histological image of better quality, above all, instead of the immunohistochemistry for s100 protein (unspecific), an image of himmunoistochemical positivity for MDM2 antibody.
Author Response
please refer to the file attached

Round 2
Reviewer 2 Report
ok for me now